# Plant Defence Mechanisms Are Modulated by the Circadian System

**DOI:** 10.3390/biology9120454

**Published:** 2020-12-09

**Authors:** Ghazala Rauf Butt, Zainab Abdul Qayyum, Matthew Alan Jones

**Affiliations:** 1Institute of Molecular, Cell and Systems Biology, University of Glasgow, Glasgow G12 8QQ, UK; 2491224b@student.gla.ac.uk; 2Department of Crop Sciences, University of Gottingen, 37073 Gottingen, Germany; zainababdul.qayyum@stud.uni-goettingen.de

**Keywords:** circadian clock, bacterial infection, plant immunity

## Abstract

**Simple Summary:**

The circadian clock is an endogenous time keeping mechanism found in living organisms and their respective pathogens. Numerous studies demonstrate that rhythms generated by this internal biological oscillator regulate and modulate most of the physiological, developmental, and biochemical processes of plants. Importantly, plant defence responses have also been shown to be modulated by the host circadian clock and vice versa. In this review we discuss the current understanding of the interactions between plant immunity and the circadian system. We also describe the possibility of pathogens directly or indirectly influencing plants’ circadian rhythms and suggest that these interactions could help us devise better disease management strategies for plants. Our review raises further research questions and we conclude that experimentation should be completed to unravel the complex mechanisms underlying interactions between plant defence and the circadian system.

**Abstract:**

Plant health is an important aspect of food security, with pathogens, pests, and herbivores all contributing to yield losses in crops. Plants’ defence against pathogens is complex and utilises several metabolic processes, including the circadian system, to coordinate their response. In this review, we examine how plants’ circadian rhythms contribute to defence mechanisms, particularly in response to bacterial pathogen attack. Circadian rhythms contribute to many aspects of the plant–pathogen interaction, although significant gaps in our understanding remain to be explored. We conclude that if these relationships are explored further, better disease management strategies could be revealed.

## 1. Introduction

Plant pathogens cause 40% of crop losses worldwide [1,2]. Viruses, fungi, nematodes, parasitic plants, and bacteria constitute the major agricultural pathogens. In spite of plants’ multi-layered defence systems, the relentless evolution of pathogens ensures a continuous “arms race” to maintain disease resistance [3,4,5]. Along with many other metabolic processes, pathogen defence is also regulated by the plant circadian system [6,7,8]. The plant circadian clock is a pervasive biological timer that contributes towards growth, development, and health [9,10]. Although there have been multiple studies examining how the plant circadian clock responds during abiotic stress, little is known about the interconnection between biotic stress and the circadian clock [11]. It is also important to acknowledge the role of light, which has an important influence on plant/pathogen interactions [12]. 

This review will examine the highlights of our understanding of the links between plant’s immune responses and circadian rhythms. Crucially, circadian rhythms regulate plants’ responses to light and so we will also consider how circadian gating of the light response influences plant/pathogen interactions [13,14]. Understanding the coordination between circadian rhythms and plants’ innate immunity will enable the development of effective mitigating strategies against diseases of particularly economically valuable crops [15,16]. 

### 1.1. Plants’ Immune and Defence System

The capacity of the plants to tolerate or to prevent a pathogen attack is described as their “innate immunity” [17]. Plants prioritize defence responses over their normal cell functions following infection by a pathogen [17,18]. Physical barriers of a plant such as its cuticle serve as the first line of defence in case of a pathogen attack. These structures prevent and avoid pathogen and pest invasion. In case of a pathogen attack, changes occur in the cuticle which are recognized by the plant and it immediately initiates defence responses [18]. 

Pathogens typically secrete Pathogen-Associated Molecular Patterns (PAMPs) when in proximity to the plant host. PAMPs are essential for pathogenicity and can be found in either physical structures or exudates such as saliva or honeydew in the case of insects [15,18]. PAMPs released by the pathogen are recognized by specialized receptor proteins known as Pattern Recognition Receptors (PRRs) present either in the plant cell/plasma membrane or within the cell in the cytoplasm. PRRs are a diverse group of proteins whose specificity is derived from the target recognised. For instance, the FLS2 PRR recognizes bacteria that produce flagellin [18]. The initial interaction between PAMPs and PRRs triggers a number of plant defence responses such as the closing of stomata to avoid pathogen invasion [15,18,19]. If the pathogen progresses within the plant cell, polymorphic Nucleotide-Binding and Leucine-Rich Repeat (NB-LRR) proteins present inside the host cell interact with specific effector molecules released by the pathogens. NB-LRRs are encoded by Resistance (R) genes in plants and confer resistance to specific pathogens [15,19]. Following this a series of defensive reactions occur starting from the production of pathogenesis-related proteins, structural cell wall changes, and synthesis of phytoalexins, concluding with the initiation of the hypersensitive response (localized cell death at the site of invasion) [20,21]. 

Phytohormones play a crucial role in plant defence, particularly Salicylic Acid (SA) and Jasmonic Acid (JA) [22]. Plants synthesize different phytohormones dependent upon pathogens’ mode of attack. For example, the SA pathway is only effective against biotrophs [17,23]. SA contributes to both local and Systemic-Acquired Resistance (SAR) and is synthesized by plants in response to pathogen attack. Plants with impaired SA signalling or synthesis are more susceptible to disease [23]. The accumulation of SA induces cell wall strengthening, ion fluxes, the production and accumulation of phenolics, and the activation of R and other defence-related genes. These responses ultimately lead to the Hypersensitive Response that results in programmed cell death [17,23,24]. 

To combat nectrotrophic pathogens, plants synthesize JA which acts via an ethylene-mediated pathway [17,22]. JA is endogenously produced, and is a conjugate between isoleucine and methyl ester which are derivates of a fatty acid class known as the jasmonates. Although the exact mechanism behind the activation of the JA pathway remains unclear, polypeptide signal molecules such as systemin and oligosaccharides hydrolysed after damages caused by the pathogens, are speculated to trigger the JA pathway into action [25]. Although the JA and SA pathways have been shown to be antagonistic, some pathogens induce both pathways [22]. 

#### Four Phases of Plants’ Immunity

There are four phases of the plant’s immune system with respect to infection stages (Figure 1). In phase 1, PRRs recognize PAMPs which restricts further colonization by the pathogen, resulting in PAMP Triggered Immunity (PTI) [19]. PTI stops the free movement of the pathogen within the host and activates defence responses irrespective of the pathogen and is a form of non-host specific resistance [15]. Later, in the second phase, pathogens produce effectors, which interact with the ongoing PTI and induce Effector-Triggered Susceptibility (ETS) [19]. ETS is specific and involves the gene-for-gene hypothesis; according to which pathogen’s virulent gene interacts with the host’s susceptibility genes causing further disease [15]. In the third phase, NB-LRR proteins particularly recognize an effector from the pathogen, causing a chain of reactions termed Effector-Triggered Immunity (ETI). ETI responses can be interpreted as an acceleration of PTI. This often results in programmed cell death around the infection site followed by disease resistance [19]. Similar to ETS, ETI is also gene-specific with plants synthesizing particular cytoplasmic R (resistance) proteins which directly interact with pathogen’s avirulent or Avr proteins [15]. The fourth phase is essentially driven by natural selection, as pathogens shed the ETI and prevent the induction of new effectors. ETI is triggered again when plants generate new NB-LRRs or gain alternative NB-LRR alleles through horizontal gene flow. This latter form of immunity acquired is described as Systemic Acquired Resistance or SAR [15,19].

### 1.2. The Plant Circadian System

The circadian clock allows plants to anticipate dusk and dawn, ultimately enabling the optimization of growth in a dynamic environment [9,10]. The optimal coordination of internal phytohormones and external cues of light and darkness is needed for the plant’s proper growth, development, and survival [26,27,28,29]. Circadian rhythms are also known to play a key role in water and carbon utilization, plants’ response towards both biotic and abiotic stresses, gas exchange and other important metabolic processes related to growth [9,26,30,31].

The plant circadian clock is an endogenous and self-sustaining biological timing mechanism [14,26,31]. There are three fundamental conditions based on which a biological rhythm can be termed as “circadian”; (1) it must continue to oscillate in the absence of environmental cues with a period of approximately 24 hours; (2) it should synchronize with or entrain light signals and temperature; (3) it should demonstrate nutritional and temperature compensation [32]. Conceptually, the plant circadian system has three major parts; 1, a central self-sustaining oscillator (clock genes), 2, the various input pathways integrating environmental cues, and 3, the output pathways [33].

Plants receive external stimuli on a daily basis to synchronize their circadian clock to prevailing environmental conditions, a process known as entrainment. A particular time or environmental cue that governs this process (such as light or temperature) is called a zeitgeber, from the German meaning “time giver” [10,12,27]. Between temperature and light, light is the major time setting mechanism for circadian clock synchronization in plants [10]. Plants entrain light signals via a network of photoreceptors that are sensitive to different portions of the solar spectrum. They can be divided into four groups; phytochromes, cryptochromes, the Zeitlupe family (ZTL), and UVR8 [10]. Phytochromes are sensitive towards red/far red-light signals while cryptochromes are sensitive towards blue light signals [27]. The ZTL family of photoreceptors similarly contribute to blue light perception [34]. Although only a small portion of the solar spectrum consists of UV-B, plants utilize the UVR8 photoreceptor to perceive these signals [35]. 

At the cellular level, the plant circadian system is comprised of individual self-sustaining circadian clocks, that are linked by inter-cellular signals (Figure 2) [31]. The molecular oscillations underlying circadian rhythms rely on a series of interconnected transcription-translation loops [27]. Numerous studies suggest that 25% to 40% of transcriptomes of Arabidopsis are under circadian regulation [26,27]. CIRCADIAN CLOCK ASSOCIATED1 (CCA1) and LATE ELONGATED HYPOCOTYL (LHY) are key transcription factors within the Arabidopsis circadian clock that are expressed in the morning. CCA1 and LHY work to suppress evening-phased factors such as *PSEUDO-RESPONSE REGULATORs*, *EARLY FLOWERING3 (ELF3)* and *GIGANTEA (GI)*. When the sun goes down, the Evening Complex or EC genes repress the expression of early *PSEUDO-RESPONSE REGULATORs*, which in turn repress the morning phased genes [36]. This feedback loop interlocks with other circadian components to generate a cellular biological rhythm with a period of approximately 24 h [13,33,36]. 

The plant circadian clock plays a critical role in physiological functions such as flowering and photosynthesis using a combination of negative and positive feedback loops. For example, mutation of GI causes a delay in flowering and a prolonged circadian period [13]. The plants’ circadian clock also interacts with and regulates methyl jasmonate, and abscisic acid pathways. Research has revealed a regulatory link between clock genes and the abundance of phytohormones involved in plant growth such as brassinosteroids, ethylene, gibberellins, and auxins [27]. The circadian rhythms are also known to play a key role in water and carbon utilization, plant’s response towards both biotic and abiotic stresses, transcription, flowering, enzyme activities, gas exchange and other important metabolic processes related to growth [9,26,31]. A study showed that wild Arabidopsis plants with a more synchronized circadian clock with their environment were fast-growing, had better survival and carbon fixation rate, and contained more chlorophyll than mutants with impaired clock functions [28].

## 2. Presence of Circadian Rhythms in Plant Pathogens

Circadian rhythms have been studied in mammals, fungi, cyanobacteria, insects, birds, and plants [32,37]. Since bacteria are unicellular organisms, typically reproducing in less than 24 h, they were initially considered not to have a circadian clock as per the “circadian-infradian rule” [38]. However, subsequent studies using cyanobacteria such as *Synechococcus* suggest a functional circadian system in photosynthetic bacteria. There have been some experiments demonstrating a rather unclear circadian network amongst the non-photosynthetic bacteria (phytopathogenic) for example in *E. coli*. However, these data are not definitive [32,38]. 

In contrast to bacteria, circadian rhythms have been well studied in fungi. Indeed, the fungal clock determines most of its life functions. The two most important concerning plant pathogenicity are sporulation and spore dispersal [39,40]. Some fungi produce and discharge their spores at night, while others at dawn or dusk. Daily rhythms in asexual reproduction have also been demonstrated. For example, in *Pilobolus spp*. spore dispersal and production are regulated by the circadian clock. The same behavior has been shown in *Pellicularia filamentosa* and *Aspergillus nidulans* [41]. The presence of circadian rhythms in these plant pathogens impacts the plant–pathogen relationship at all infection stages [39]. The fungus Neurospora has been studied widely to understand the fungal circadian clock and serves as a model fungal circadian system [40]. 

Although there have been numerous studies on the presence of a circadian clock in *Caenorhabditis elegans*, a free-living soil-inhabiting nematode, not much is known about the definite presence of circadian clock in the plant pathogenic nematodes [42,43]. In fact, a study revealed the absence of photoreceptor homologs of *C. elegans* in the phytopathogenic root-knot nematodes, suggesting that they cannot perceive light signals and do not demonstrate circadian rhythms. They also do not show any biological rhythmicity during any stage of their life such as egg hatch etc [42,43,44]. However, there have also been studies indicating that root-knot nematodes are influenced by the plants’ circadian clock. As plants become comparatively susceptible towards pathogens at night, nematodes perceive these signals and penetrated plant host cells efficiently when inoculated during night hours [42]. 

*Drosophila melanogaster’s* circadian clock is used as a model in insect systems and is one of the best understood circadian mechanisms. The Drosophila clock contains feedback loops and transcription–translation interlinks, which function together to synchronize daily behavior with the external environment [37,45]. Importantly, Haematophagous insects’ feeding activities are regulated by the circadian clock. Since insects from this class are vectors of many plant diseases, their clock regulation of feeding is critical for pathogen transmission [37,46]. 

## 3. Modulation of Plant’s Immune Responses by the Circadian Clock 

Scientific evidence is growing in support of the regulation of plant–pathogen interaction by the circadian clocks of both the pathogen and the plant. Studies supporting that the plant immunity indeed is regulated by its circadian clock are briefly documented here [8,16]. A microarray data study concluded that many genes responsible for PTI or PAMP-triggered immunity, like FLS2, express rhythmically in Arabidopsis. Pathogen responsive genes, such as glycine-rich RNA binding protein (GRP), found in barley and many other crops also show circadian regulation [8]. GRP directly binds with PAMPs released by the pathogen, thus enabling the plant to recognize the pathogen and initiate PTI [47]. When attacked by a fungal pathogen, two GRPs namely HvGRP2 and HvGRP3 of barley were expressed at a higher rate, their respective levels varied when barley was placed in a light/dark cycle of 16/8 h respectively. Likewise, in Arabidopsis, AtGRP7 (also a member of GRP) was shown to influence the stomatal opening, response towards stresses and flowering and proved to be regulated by the circadian clock. It has been shown to work with CCA and LHY to control the stomatal defence response of plants [8]. 

The major defence mechanism of plants against specific pathogens is the R (resistance) gene system, which has also been shown to be influenced by the circadian system. Impaired clock mutants of Arabidopsis show a defect in the R gene and even basal resistance [48]. For instance, Goodspeed et al. [49] experimented on Arabidopsis plants attacked by an herbivore (cabbage loopers). They concluded that cabbage loopers almost always prefer to feed on arrhythmic plants proving that the circadian clock increases resistance towards herbivore attack mainly through affecting the SA and JA pathways [50]. One of the six acyl-coa-binding proteins (ACBP) in Arabidopsis, ACBP3, was studied to better understand its role in the resistance and plant defence; the same protein was found to be regulated by the circadian clock [51]. 

The morning phased *CCA1* and *LHY* genes positively regulate plant resistance against oomycetes and bacterial pathogens like *P.syringae*. CCA1 regulates the plant–pathogen interaction by contributing to resistance responses. Plants lacking *CCA1* show increased susceptibility at evening while being highly resistant in the morning. These rhythmic susceptibilities during the course of 24 hours were not visible in *CCA1-ox* mutants, thus indicating a relationship between CCA1 and plant immunity [52]. Similarly, resistance against downy mildew is impaired in *cca1* seedlings whereas overexpression of *CCA1* led to improved resistance [5]. Characterization of the tomato gene *DEA1* (which is expressed upon infection by the late blight pathogen *Phytophthora infestans*) showed that *DEA1* is modulated by both the circadian clock and light [53]. Some experiments conducted on plant defence pathways via stomata suggest that clock genes control resistance towards bacterial pathogens through stomatal opening timing. For example, Arabidopsis showed resistance towards *P.syringae* at night. However, plants can close stomata actively during bacterial invasion to restrict entry upon PAMP recognition. Crucially, *CCA1* and *LHY,* both regulate this gating response of stomatal opening and closing [8].

The interplay between the circadian system, light, and pathogen resistance has also been explored. One of the studies explored the links between red light and resistance showed by Arabidopsis towards *P. syringae pv*. *tomato* DC300. Plants showed increased susceptibility just before midnight. RNA-seq analysis showed that red light-triggered resistance responses regulated by the circadian clock and therefore, increased the chance of survival against pathogens. Furthermore, it was revealed that the circadian regulated genes interacted with various plant hormones, phytochromes, and induced the SA mediated defence responses [54]. Another study conducted on comparing the activity of flagellin-sensing2 (a PRR that recognizes bacterial flagellin) in wild type and arrhythmic mutants of Arabidopsis plants concluded that when infected with *Pseudomonas syringae* in the morning the expression of FLS2 was stronger in the wild type and not in the mutants [13,55]

Interestingly, some studies have revealed that plants anticipate possible infection using their circadian clocks [48]. Time of the day and circadian rhythms directly play a role in the functionality of host’s immune system and therefore they also affect the virulence of pathogens, intensity of infection, colonization and damage to host cells, and the overall outcome of host–pathogen relationship [29]. The circadian clock in a way decides the most appropriate time of the day for efficient plant immune responses [52]. Plants show enhanced resistance towards pathogens during daytime hours compared to the night. Importantly, plants’ defence system exhibits circadian oscillations even in the absence of a pathogen [42]. For instance, Arabidopsis infected by *P. syringae* DC3000 showed increased tolerance towards infection at certain times of the subject day, i.e., in the morning. These temporal regulations help plants to not only better respond but anticipate when the next infection may occur [56]. 

Conversely, pathogen–plant interactions can often reset the plant’s circadian clock, resulting in the reallocation of the limited resources used for the development and growth of the plant [8]. For example, experiments using *Paulownia fortunei* concluded that circadian gene expression was altered when infected with the Paulownia Witches’ Broom phytoplasma [6]. Similarly, pathogens can manipulate hormone signaling thus altering the plant’s circadian system. Effectors of *P. syringae* cause the production of abscisic acid and auxins, both of which are regulated by the clock and which can also regulate clock function. SA and JA signaling can also be manipulated by pathogens [8]. Importantly, minor alteration in clock genes can cause a change in the plant’s defence responses [36]. Following a very localized infection at a single leaf in Arabidopsis, it was observed that the amplitude of the circadian clock slowed down and period length increased even in the distant un-infected tissues [11]. When treated with the defence phytohormone SA, the same results were noticed [11]. Another study completed using susceptible eds4 Arabidopsis concluded that *eds4* seedlings showed altered clock responses compared to the wild type Arabidopsis plants [57]. *eds4* seedlings were less sensitive towards the red and white light, their flowering time was accelerated, and their leaf movement had a longer period as compared to wild type controls. Other circadian responses and clock profiles were also altered in the *eds4* plants. The same study found that a bacterial infection induces substantial reconfiguration in the circadian clock genes expression for example downgrading of the morning phased genes, resulting in an increase of bacterial infection and susceptibility of host plant [57].

## 4. Interactions between the Circadian System, Light, and Plant Defence

Light has a crucial role in both plant growth and response towards pathogen attack, whilst also serving as one of the primary zeitgebers that entrain the circadian system [10,58]. The regulation of plants’ defence by light and the circadian system allows plants to anticipate periods of likely infection and thus periodically increase their resistance [12]. Although additional work is needed to understand these interactions—particularly in natural or field settings—it is of interest to discuss the interplay between light, circadian signalling, and plants’ immune response [12].

Light and the circadian system both contribute to innate immunity—in part by contributing to the maintenance of the physical barriers that restrict pathogen ingression. For example, stomatal opening is regulated by both light and the clock. Inoculation with pathogenic bacteria via direct infiltration (bypassing stomata) negates the contribution of the circadian system to innate immunity, although light maintained a significant contribution [12,20,58].

Plants show attenuated defence responses towards viral, fungal, and bacterial pathogens when grown in the dark [8,54]. Indeed, a direct link between light and plant–pathogen compatibility is becoming more evident in studies particularly regarding SA and other defence mechanisms of plants [12]. Light quality, fluence rate, and duration influence plants’ immunity and defence by regulating the sensitivity of plants towards SA, development of the HR, and expression of pathogenesis-related genes [5,20,59]. Inoculation with *Pseudomonas syringae pv. maculicola,* during the day, induces a more substantial response compared to plants inoculated at night. The dependence of the SA signalling pathway on light irradiation is one possible explanation of these data [5]. Similarly, the JA pathway is also modulated by light. Red and Far-Red (or the R/FR ratio) regulate the activation of JA. *phyB* mutants, which have impaired red/far-red responses, were found to be more prone to infection caused by the fungus *Fusarium oxysporum* [5]. 

In some cases, the chloroplast electron transport chain drives the production and physiological functions of several reactive oxygen species (ROS) that contribute to plant defence [60]. Rapid production of ROS is a first-line defence response, and the interplay between cellular redox state and the circadian clock ensures an equilibrium between plant growth and immunity [39,52]. Importantly, Arabidopsis mutants lacking nonphotochemical quenching also lack PTI [5]. When infected with avirulent strains of *Pseudomonas syringae*, Arabidopsis plants maintained in constant darkness showed increased bacterial infection and decreased resistance in comparison with Arabidopsis plants grown in the presence of light and infected with the same bacteria [60]. Impaired photosystem function and reduced light input similarly have a positive impact on the susceptibility of *Nicotiana benthamiana* towards the Turnip Mosaic Virus (TVC) [5]. 

HR and ETI particularly require light signalling for activation [12,58]. Arabidopsis infected with TVC and grown in the dark showed a reduced HR and a suppressed resistance in comparison to the ones treated in the light [61]. The study of the TVC pathosystem revealed that photoreceptors cryptochrome1, cryptochrome2, phototropin1, and phototropin2 are required for mediation of HR conferred by R gene termed as HRT [5]. Various experiments using maize, transgenic tomatoes, rice, and Arabidopsis mutants hint at a link between light and HR [58]. For example, phytochromes are thought to play a role in PRR gene expression. Plants with the varying activity of phytochrome A and B were grown in darkness and high fluence white light respectively. Those which were grown in darkness had no expression of HR and PRR when treated with SA, while those in the light had a proper expression. Plants lacking phytochromes A and B similarly demonstrated complete loss of PRR expression, indicating their role in this important defence signalling pathway [20]. In addition, the blue light photoreceptor cryptochrome1 is required for defence against bacterial infection only under continuous light and not when the plant is under short daylight conditions [5].To test this hypothesis further, Arabidopsis was infected with *Pseudomonas syringae*, the results demonstrated a clear connection between the accumulation of SA and the presence of light [12].

## 5. Conclusions 

A successful plant pathogen infection requires a virulent pathogen, a susceptible host, and environmental conditions favouring the pathogen [12]. Through various experimentation, it has been established that plants’ innate immunity involves crosstalk across multiple pathways including light signalling and the circadian clock [62]. As circadian rhythms have beneficial effects upon physiological, developmental and biochemical processes of living organisms, it is apparent that they also influence drug efficiency and disease treatments. Various mechanisms have been hypothesized on how the circadian system contributes to plants’ immunity and defence as both adaptive and innate immunities are modulated by the circadian system [58]. Unravelling the mechanisms underlying the relationship between the circadian system and plant immunity carries great importance for plant health and disease management [63,64]. The emerging field of chrono-immunotherapy (synchronizing time of medicine with the circadian clock of the body to optimize treatment) demonstrates the utility of this approach in treating disease [63]. As a consequence, efforts to improve the efficiency of plant defense responses using targeted interventions within the circadian system should be explored further. Our understanding of the circadian clock controlling the plant defense in different ways provides a foundation for future work [64,65,66]. The synchronization of disease treatment and plants’ internal clock could result in efficient disease control and decreased crop yield losses [64,67]. 

## Figures and Tables

**Figure 1 biology-09-00454-f001:**
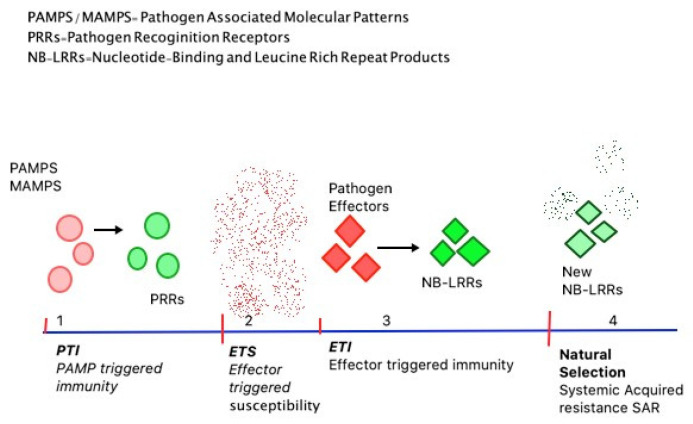
The four phases of plant immunity; (1) PAMPS or Pathogen Associated Molecular Patterns released by the pathogens come in contact with the plant’s PRRs or Pathogen Recognition Receptors used by the plants to recognize pathogens triggering PAMP Triggered Immunity (PTI) or PAMP triggered immunity. (2) Pathogen progresses inside the plant cell taking over PTI and causing ETS or Effector-triggered susceptibility. (3) Plant Nucleotide-Binding and Leucine-Rich Repeat Products (NB-LRRs) interact with pathogen effectors initiating ETI (effector-triggered immunity) responses by the plant. (4) Pathogens produce new effectors proving ETI insufficient. At this stage, natural selection plays its role, plants develop new resistance genes over the course known as Systemic Acquired Resistance or the pathogen invasion is successful [15,19].

**Figure 2 biology-09-00454-f002:**
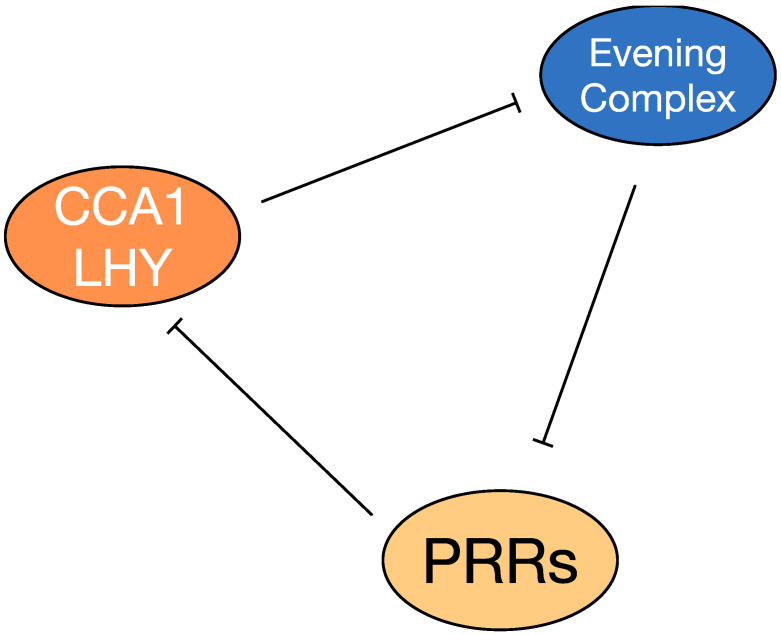
An oversimplified model of the Arabidopsis circadian system. CCA1/LHY are induced at dawn and repress expression of the Evening Complex, which represses itself while also repressing expression of *PRR* genes. Accumulation of PSEUDO-RESPONSE REGULATORs (PRRs that are distinct from Pattern Recognition Receptors) represses numerous genes, including *CCA1* and *LHY*. More detailed descriptions of the circadian system are detailed elsewhere [13,33,36].

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
