# Peer review of "Plant Defence Mechanisms Are Modulated by the Circadian System"

_biology, 2020, doi:10.3390/biology9120454_

Round 1

Reviewer 1 Report

Good job. I appreciate the authors for their good work. 

Out of curiosity I just wanted to know, In biotic stress and the circadian clock interactions. the circadian clock of the pathogen or the plant, who's  circadian clock plays a dominant role in the interaction and why? Also in the interaction of the soil pathogen with the plant roots do darkness existing in the soil rhizosphere favors the infection? 

I suggest the authors can check the consistency of writing the scientific names and the gene name in the manuscript following the nomenclature.

Thank you 

Author Response

We thank the reviewers for their considered comments, which we address in full below. We have included our responses in bold text.

Out of curiosity I just wanted to know, In biotic stress and the circadian clock interactions. the circadian clock of the pathogen or the plant, who's  circadian clock plays a dominant role in the interaction and why?

This is a very interesting question. However, it depends on many known and unknown factors such as the pathogen type (fungi, nematode, bacteria) and if circadian clock is present in them or not? In general, we would assume that the plants’ clock is predominant in governing an appropriate response, although additional work will be required to assess this hypothesis.

Also in the interaction of the soil pathogen with the plant roots do darkness existing in the soil rhizosphere favors the infection?

Although living in near-darkness, pathogens in the rhizosphere will still experience daily rhythms of temperature change and nutrient availability (from plant-derived photosynthates being transported to the soil). This is a very interesting question, although additional work will be required to address it.

I suggest the authors can check the consistency of writing the scientific names and the gene name in the manuscript following the nomenclature.

We have revised formatting as requested.

Reviewer 2 Report

It is a nice review describing our current understanding about circadian rhythms and the role it plays in modulating plant immunity (both adaptive and innate immunity)  especially plant/pathogen interactions. It also provides a great deal of information on plant immunity, defense systems, phases of plant immunity, circadian rhythms in pathogens and importance of interactions happening between circadian clock , light and plant defense. The review is well-written and well organized. The review will advance the reader’s knowledge about circadian clocks and its role not only in regulating plant immunity but also in modulating growth in dynamic environments and responding to abiotic and biotic stresses.

Although this review briefly discussed the importance of synchronization of disease treatments and circadian clock in efficient disease control. However, the review does not describe how this knowledge can be implemented in practical disease control. It is recommended that author’s include some literature in the role of circadian clock and practical disease control.

Author Response

We thank the reviewers for their considered comments, which we address in full below. We have included our responses in bold text.

Although this review briefly discussed the importance of synchronization of disease treatments and circadian clock in efficient disease control. However, the review does not describe how this knowledge can be implemented in practical disease control. It is recommended that authors include some literature in the role of circadian clock and practical disease control.

We thank the reviewer for this comment. As this is a relatively new field, we are unaware of examples in the literature in which the clock has been successfully modulated to enhance disease control- this is principally because the clock can be ‘driven’ by environmental cycles of light and temperature, and so precisely manipulating the clock in natural conditions is difficult. We propose that circadian timing could be temporarily shifted via drug application although we are still completing experiments to achieve this. We have modified our conclusion to emphasise the potential benefits of this approach.